# Reliability of the Garden Alignment Index and Valgus Tilt Measurement for Nondisplaced Femoral Neck Fractures

**DOI:** 10.3390/jpm13010053

**Published:** 2022-12-27

**Authors:** Yasuaki Yamakawa, Norio Yamamoto, Yosuke Tomita, Ryuichiro Okuda, Yasutaka Masada, Akihiro Shiroshita, Toshiyuki Matsumoto

**Affiliations:** 1Department of Orthopedic Surgery, Kochi Health Sciences Center, Kochi 781-8555, Japan; 2Department of Orthopedic Surgery, Miyamoto Orthopedic Hospital, Okayama 773-8236, Japan; 3Department of Epidemiology, Graduate School of Medicine, Dentistry and Pharmaceutical Sciences, Okayama University, Okayama 700-8558, Japan; 4Scientific Research Works Peer Support Group (SRWS-PSG), Osaka 541-0043, Japan; 5Department of Physical Therapy, Faculty of Health Care, Takasaki University of Health and Welfare, Gunma 370-0033, Japan; 6Division of Epidemiology, Department of Medicine, Vanderbilt University School of Medicine, 2525 West End Avenue, Nashville, TN 37203, USA; 7Division of Allergy, Pulmonary and Critical Care Medicine, Department of Medicine, Vanderbilt University School of Medicine, Nashville, TN 37232, USA

**Keywords:** femoral neck fracture, intracapsular hip fracture, Garden alignment index, posterior tilt, inter-rater reliability, intra-rater reliability, intraclass correlation coefficients

## Abstract

Anteroposterior (AP) alignment assessment for nondisplaced femoral neck fractures is important for determining the treatment strategy and predicting postoperative outcomes. AP alignment is generally measured using the Garden alignment index (GAI). However, its reliability remains unknown. We compared the reliability of GAI and a new AP alignment measurement (valgus tilt measurement [VTM]) using preoperative AP radiographs of nondisplaced femoral neck fractures. The study was designed as an intra- and inter-rater reliability analysis. The raters were four trauma surgeons who assessed 50 images twice. The main outcome was the intraclass correlation coefficient (ICC). To calculate intra- and inter-rater reliability, we used a mixed-effects model considering rater, patient, and time. The overall ICC (95% CI) of GAI and VTM for intra-rater reliability was 0.92 (0.89–0.94) and 0.86 (0.82–0.89), respectively. The overall ICC of GAI and VTM for inter-rater reliability was 0.92 (0.89–0.95), and 0.85 (0.81–0.88), respectively. The intra- and inter-rater reliability of GAI was higher in patients aged <80 years than in patients aged ≥80 years. Our results showed that GAI is a more reliable measurement method than VTM, although both are reliable. Variations in patient age should be considered in GAI measurements.

## 1. Introduction

The incidence of hip fractures, including femoral neck fractures, has increased [1]. Failed internal fixation for femoral neck fractures has a strong negative impact on patients, leading to increased postoperative mortality and high medical costs [2]. Recent systematic reviews and meta-analyses have revealed that preoperative posterior tilt ≥ 20° using lateral radiograph is associated with failed internal fixations [3,4]. On the other hand, preoperative anteroposterior (AP) alignment (valgus tilt > 15°) was a risk factor for failure in treatment [5]. In addition, the influence of postoperative AP alignment on reoperations or functional scores was evaluated recently [6,7].

Posterior tilt measurement using lateral radiography was first presented as an assessment of lateral alignment in 2009 [8]. The reliability of the measurement ranges from substantial to excellent [9,10,11]. On the other hand, AP alignment is generally measured using the Garden alignment index (GAI) [12], but its reliability remains unknown. It is challenging to determine alignment by the trabecular line in the femoral head [12], especially in elderly patients with thin trabecular lines because of osteoporosis. Therefore, surgeons need an AP alignment measurement with high reliability to develop treatment strategies and estimate the prognosis. We hypothesized that valgus tilt measurement (VTM) using an AP radiograph would be as reliable as posterior tilt measurement using a lateral radiograph.

The purpose of this study was to evaluate and compare the reliability of GAI and a new AP alignment measurement (VTM) using preoperative AP radiographs of nondisplaced femoral neck fractures. In addition, we compared the reliability in terms of raters’ status (junior vs. senior surgeons) and patient age (over vs. under 80 years of age). We believe that determining a more reliable measurement for AP alignment will contribute to better clinical decision making and could serve as a basis for extensive clinical research on femoral neck fractures.

## 2. Materials and Methods

### 2.1. Study Design and Setting

The study was designed as an intra- and inter-rater reliability analysis in a general hospital. We followed the standards of earlier reliability studies [13,14,15]. This study was conducted in accordance with the Declaration of Helsinki and was approved by the institutional review board (approval number: 221043).

### 2.2. Patient Selection

We calculated the sample size for reliability measurements using a web calculator [16]. The sample size was a total of 16 images based on data (expected intraclass coefficient [ICC] of inter-rater reliability = 0.8, precision [±expected] = 0.1, number of raters = 4, α = 0.05) reported in reliability studies [9,17]. Lastly, we set a sample size of 50 images because earlier reliability studies evaluated 50 images [9,11].

We selected consecutive patients with nondisplaced femoral neck fractures (Garden stage I and II) according to the Garden classification on preoperative AP radiographs between March 2019 and June 2022 [12]. We only excluded radiographs in which we could not measure VTM or GAI for any reason in order to avoid selection bias and in consideration of the clinical practice setting (Figure 1).

### 2.3. AP Hip Radiographs

Radiology technicians obtained preoperative AP hip radiographs using standard methods. The patients were placed in the supine position. If the fractured leg was externally rotated, the leg was manually positioned in its natural position to the extent possible.

### 2.4. Radiographic Measurements

The raters were four trauma surgeons (two junior and two senior surgeons) in the same general hospital. We defined the raters as senior surgeons (more experienced surgeons with 10 years of experience) or junior surgeons (less experienced surgeons with <10 years of experience). All raters were lectured on the measurements of GAI and VTM. The raters practiced the measurements using 10 radiographs as pilot measurements before the start of the study. The raters used a digital Picture Archiving and Communication System (PACS) with standard resolution monitors, and variables such as patient age and sex were noted.

First, all raters independently measured the angles on both the fractured and the unfractured sides in all 50 images. After a washout period of 6 weeks [9,10], the order of images at the time of the second viewing was randomly changed. They measured the angles in the same image set for a second time.

### 2.5. Garden Alignment Index

GAI is the angle between the trabecular line in the femoral head and a line drawn through the long axis of the medial cortex of the femoral shaft [12] (Figure 2). A valgus displacement of 15° based on GAI is the cutoff value for valgus-impacted femoral neck fracture [5,18]. The valgus displacement is the difference in angles measured with GAI between the fractured side and the unfractured side in valgus-impacted femoral neck fractures.

### 2.6. Valgus Tilt Measurement

The valgus tilt was measured using AP radiographs with modified methods of posterior tilt measurements using lateral radiographs [8]. First, the mid-neck line (MNL) was drawn through the center of two lines across the residual mid-femoral neck; the first line was drawn at the narrowest part of the residual mid-femoral neck, and a second parallel line was drawn 5 mm distal to the first line. Second, the femoral head line (FHL) was drawn from the center of the femoral head circle to the point where the MNL crosses the femoral head circle. Lastly, valgus tilt was the angle formed by the MNL and FHL (Figure 3). Negative values denoted a varus tilt of the femoral head corresponding to the MNL, whereas positive values denoted a valgus tilt.

### 2.7. Statistical Analysis

Intra-rater reliability reflects the variation in measurements by a single rater across multiple observations, while inter-rater reliability reflects the variation in measurements between multiple raters [15]. The ICC was used to evaluate the intra- and inter-rater reliabilities of GAI and VTM. To calculate intra-rater reliability, we used a mixed-effects model, considering rater as a fixed effect, and patient and time as random effects. The ICC for intra-rater variability was calculated on the basis of the mean rating of four raters and the absolute agreement. To evaluate inter-rater reliability, we used a mixed-effects model considering time as a fixed effect, and rater and patient as random effects. The ICC for inter-rater variability was calculated on the basis of the mean rating of two timepoints and absolute agreement. The 95% confidence intervals (CI) were calculated using bootstrap resampling methods (1000 replications). We interpreted the ICC as follows according to a previous study [19]: excellent (>0.75), fair to good (0.40–0.75), and poor (<0.40).

For inter-rater reliability, the standard error of measurement (SEM agreement) was calculated from the sum of raters and residual variance: SEM agreement = √(σ between raters + σ residual) [20]. The minimal detectable change (MDC) was calculated as 1.96 × √2 × SEM. For intra-rater reliability, the within-subject standard deviation (SD) and repeatability coefficient (RC) were calculated. The within-subject SD was calculated using one-way analysis of variance. RC was calculated as √2 × 1.96 × within-subject SD [21].

ICCs for intra- and inter-rater reliabilities were compared between GAI and VTM using bootstrap resampling methods. Inter- and inter-rater reliabilities were compared between GAI and VTM using descriptive statistics. We also compared the reliability of GAI and VTM between senior and junior doctors.

In the subgroup analysis, we compared the inter- and intra-rater reliability of GAI and VTM in four raters between two patient groups (patients aged ≥80 years vs. <80 years).

Using four measurements for each case at the first test session, we calculated the degree using VTM as a reference, the degree of the unfractured side, and the degree corresponding to a valgus displacement of 15° as the cutoff value based on GAI.

Statistical analysis was performed using R version 4.2.2 (R Foundation for Statistical Computing, Vienna, Austria).

## 3. Results

We excluded one radiograph in which we could not measure VTM because of excessive external rotation (Figure 4). Finally, this study included 50 patients (39 [78%] women and 11 [22%] men), with a median age of 78 (IQR 38–99) years. Twenty-four and 26 patients were aged ≥80 years and <80 years, respectively. Four raters measured GAI and VTM in 50 hip radiographs in two tests, providing a total of 4 × 50 × 2 = 400 assessments.

### 3.1. Intra-Rater Reliability

The overall ICC for the four raters was “excellent” for both GAI (ICC 0.92, 95% CI 0.89–0.94) and VTM (ICC 0.86, 95% CI 0.82–0.89) (Table 1). The difference in ICC between GAI and VTM was 0.08 (95% CI, 0.03–0.14). The within-subject SD and RC of VTM were lower (4.92, 13.65) than those of GAI (6.33, 17.54). The inter-rater reliability of junior surgeons was similar to that of senior surgeons in GAI, but was higher than that of senior surgeons in VTM.

### 3.2. Inter-Rater Reliability

The overall ICC for the four raters was “excellent” for both GAI (ICC 0.92, 95% CI 0.89–0.95) and VTM (ICC 0.85, 95% CI 0.81–0.88) (Table 2). The difference in ICC between GAI and VTM was 0.08 (95% CI 0.03–0.13). The SEM and MDC values of GAI were lower (2.35 and 6.51) than those of VTM (2.56 and 7.08). The inter-rater reliability of junior surgeons was similar to that of senior surgeons in GAI but was higher than that of senior surgeons in VTM.

### 3.3. Subgroup Analysis

In the subgroup analysis, the intra- and inter-rater reliabilities of GAI and VTM were higher in patients aged <80 years than in patients aged ≥80 years (Table 3).

The mean degree of VTM on the unfractured side was mean 1.4° (SD 4.3) and the median was 2° (interquartile range [IQR] 1–4). The mean degree of VTM corresponding to a valgus displacement of 15° based on GAI was 9.3° (SD 6.4), with a median of 7° (IQR 5–15).

## 4. Discussion

The results demonstrated that GAI was a more reliable measurement method than VTM in assessing AP alignment using preoperative AP radiographs for nondisplaced femoral neck fractures, although both measurements were reliable. The reliability of junior surgeons was similar to that of senior surgeons for GAI, but was higher than that of senior surgeons for VTM. The reliability of GAI in patients aged <80 years was higher than that in patients aged ≥80 years.

GAI is a more reliable measurement method than VTM. The inter- and intra-rater reliability of the GAI was higher than that of the posterior tilt assessment using lateral radiographs (both ICC 0.77) [9], although the analysis methods were different from our methods. The results of the assessment using AP radiographs were not consistent with those using lateral radiographs (inter- and intra-rater reliability of posterior tilt [ICC 0.75] versus inter- and intra-rater reliability of lateral GAI [ICC 0.60, 0.75]) [11], suggesting that the reliability of posterior tilt was higher than that of lateral GAI. In the AP alignment assessment, our results objectively demonstrated the reasons for the historically frequent use of GAI in terms of reliability.

The reliability of junior surgeons was similar to that of senior surgeons for GAI but higher than that of senior surgeons for VTM. The results differed from general expectations because the reliability of experienced raters is usually higher than that of inexperienced raters [14]. We consider GAI to be a reliable measurement method regardless of experience because GAI was similar between junior and senior surgeons. In contrast, in VTM, junior surgeons seemed to consistently identify the narrowest part of the residual mid-femoral neck. Thus, junior surgeons may be superior in terms of assessments based on morphological indicators.

The reliabilities of GAI and VTM were higher for patients aged <80 years than for those aged ≥80 years. The view of the trabeculae affects GAI measurement because it is measured using the trabecular line in the femoral head. It is challenging to determine alignment using the trabecular line for radiographs of elderly patients with osteoporosis. On the other hand, MNL in VTM may not be a constant parameter due to age-related degeneration. It is important to recognize that measuring AP alignment is not as reliable in older patients as it is in younger persons.

### 4.1. Strengths

This is the first study to evaluate the reliability of GAI assessment. Considering the evidence from previous studies [8,9,10,11], we compared the reliability of GAI with that of VTM as a new measurement scope. We followed the standards of earlier reliability studies [13,14,15] and chose not to exclude AP radiographs of poor quality to minimize the risk of selection bias.

### 4.2. Limitations

This study had some limitations. First, the rotation and flexion of the injured hip due to pain might have affected AP alignment measurements, although the influence on posterior tilt assessment with lateral radiographs was negligible for positions of the injured hip [22]. Second, there was a lack of external validity because all the data were obtained from only one general hospital in Japan. It is unclear whether the results of this study can be generalized to other countries with different patient populations and image viewing systems. Ultimately, well-designed studies with more images and raters are necessary to clarify the reliability of AP alignment assessment.

## 5. Conclusions

This reliability analysis showed that although both GAI and VTM were reliable measurement methods, GAI was more reliable than VTM for assessing AP alignment using preoperative AP radiographs for nondisplaced femoral neck fractures. Additionally, the reliability of GAI was higher in patients aged <80 years than in patients aged ≥80 years. Therefore, age-related variations in GAI measurement should be considered. Well-designed reliability studies with more images and raters are necessary to clarify the reliability of AP alignment assessment.

## Figures and Tables

**Figure 1 jpm-13-00053-f001:**
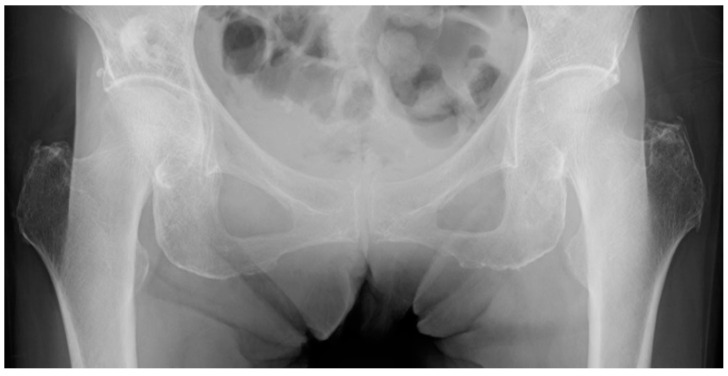
Preoperative anteroposterior radiograph showing a left nondisplaced femoral neck fracture. The patient was a 71 year old woman of 140 cm height and 34 kg weight (body mass index 17.3 kg/m^2^). We did not exclude the images that were difficult to measure due to the influence of soft tissues or radiation dose.

**Figure 2 jpm-13-00053-f002:**
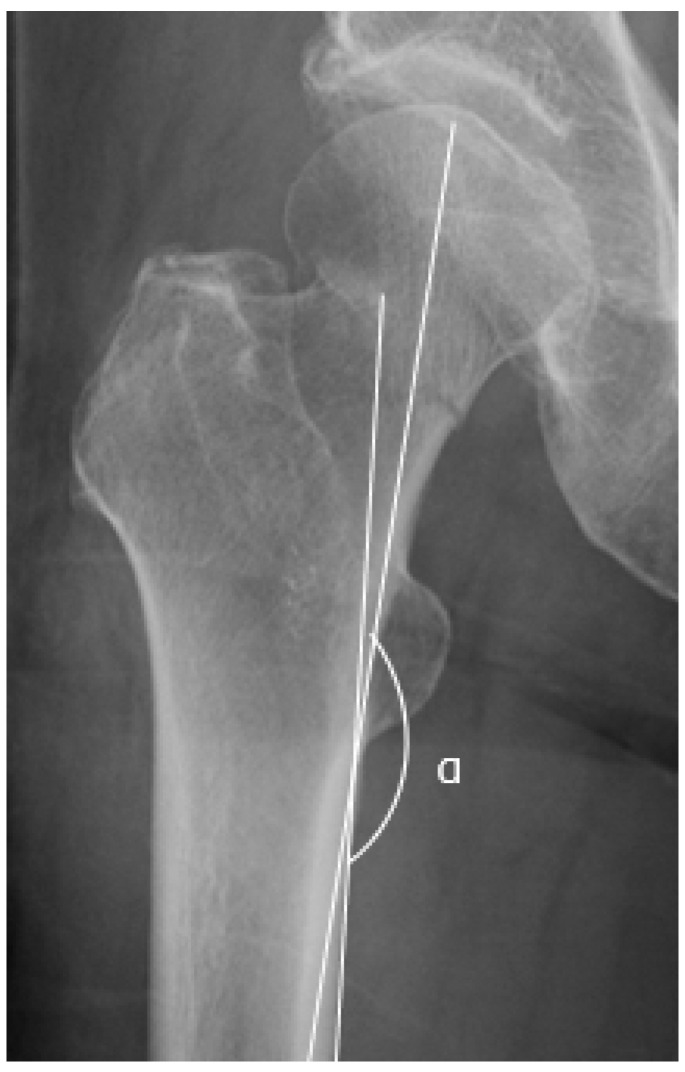
Garden alignment index using a preoperative anteroposterior radiograph. The angle of GAI is denoted by α.

**Figure 3 jpm-13-00053-f003:**
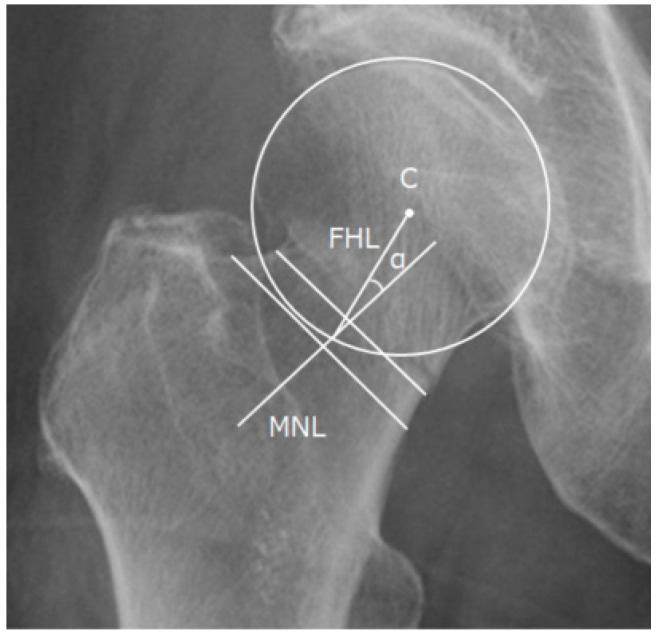
Valgus tilt measurement using preoperative anteroposterior radiographs. First, the mid-neck line (MNL) was drawn through the center of two lines across the residual mid-femoral neck; the first line was drawn at the narrowest part of the residual mid-femoral neck, and a second parallel line was drawn 5 mm distal to the first line. Second, the femoral head line (FHL) was drawn from the center (C) of the femoral head circle to the point where the MNL crossed the femoral head circle. Lastly, the valgus tilt (α) was the angle formed by the MNL and FHL.

**Figure 4 jpm-13-00053-f004:**
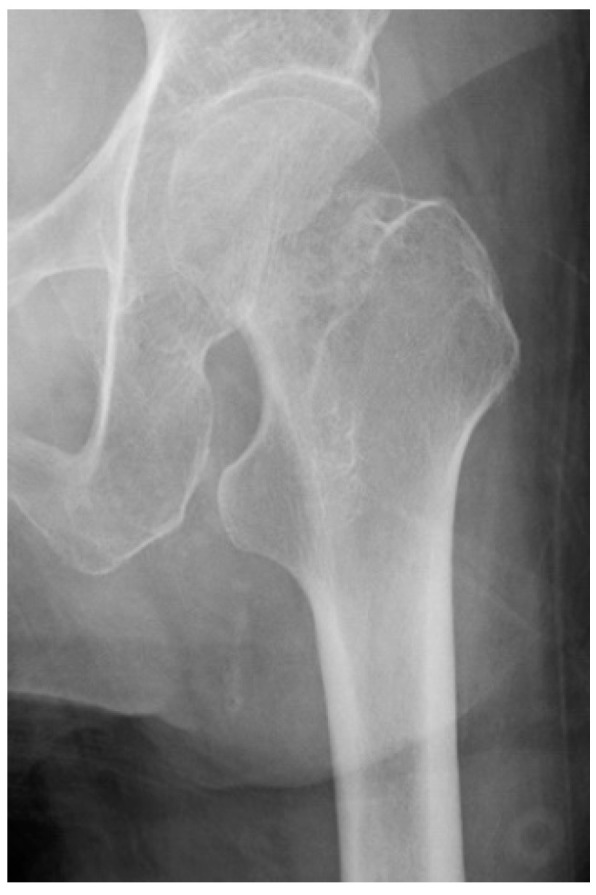
Image of measurement difficulties by valgus tilt measurement.

**Table 1 jpm-13-00053-t001:** Intra-rater reliability of the Garden alignment index and valgus tilt measurement for four raters in 50 hip radiographs.

	Garden Alignment Index	Valgus Tilt Measurement	Difference
	ICC (95% CI)	ICC (95% CI)	Point Estimate (95% CI)
Four raters	0.92 (0.89–0.94)	0.86 (0.82–0.89)	0.08 (0.03–0.14)
Two senior surgeons	0.94 (0.90–0.97)	0.88 (0.82–0.93)	
Two junior surgeons	0.94 (0.90–0.97)	0.94 (0.90–0.97)	

ICC, intraclass correlation coefficient; CI, confidence interval.

**Table 2 jpm-13-00053-t002:** Inter-rater reliability of the Garden alignment index and valgus tilt measurement for four raters in 50 hip radiographs.

	Garden Alignment Index	Valgus Tilt Measurement	Difference
	ICC (95% CI)	ICC (95% CI)	Point Estimate (95% CI)
Four raters	0.92 (0.89–0.95)	0.85 (0.81–0.88)	0.08 (0.03–0.13)
Two senior surgeons	0.95 (0.91–0.98)	0.90 (0.83–0.94)	
Two junior surgeons	0.95 (0.91–0.97)	0.95 (0.91–0.97)	

ICC, intraclass correlation coefficient; CI, confidence interval.

**Table 3 jpm-13-00053-t003:** Intra- and inter-rater reliability of the Garden alignment index and valgus tilt measurement for four raters in patients aged ≥80 years vs. aged <80 years.

Patients	Measurement	Intra-Rater Reliability ICC (95% CI)	Inter-Rater Reliability ICC (95% CI)
Patients aged <80 years	GAI	0.93 (0.91–0.95)	0.95 (0.92–0.98)
	VTM	0.87 (0.82–0.91)	0.88 (0.82–0.92)
Patients aged ≥80 years	GAI	0.88 (0.82–0.93)	0.90 (0.84–0.95)
	VTM	0.83 (0.77–0.88)	0.80 (0.73–0.85)

GAI, Garden alignment index; VTM, valgus tilt measurement; ICC, intraclass correlation coefficient; CI, confidence interval.

## Data Availability

The data supporting the findings of this study are available from the first author, Y.Y., upon reasonable request.

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
