# Peer review of "Reliability of the Garden Alignment Index and Valgus Tilt Measurement for Nondisplaced Femoral Neck Fractures"

_jpm, 2022, doi:10.3390/jpm13010053_

Round 1

Reviewer 1 Report

Dear Authors

The work submitted for review is interesting, but requires additions before it can possibly be accepted for publication. The title is too long-please shorten it. The keywords are too descriptive, please shorten to 5-6 entries maximum. The introduction section is currently unsuitable for publication. It is firstly too generally written, and secondly it is rather a part of the introduction to another section-Discussion. Please enlarge this part of the paper to make it a real introduction to the topic of your research. I would suggest replacing the phrase "junior and  senior surgeons" in the paper with surgeon with seniority.... and a surgeon with seniority.....
The paper should be enriched with more example photos. The Conclusion section should be completed, its current form is too laconic.

Regards

Author Response

Response to Reviewer 1 Comments:

The work submitted for review is interesting, but requires additions before it can possibly be accepted for publication. 

Response: We thank Reviewer 1 for these comment.

The title is too long-please shorten it. 

Response: We agree with you and have shortened it to:

Line 2: Reliability of the Garden alignment index and valgus tilt measurement for nondisplaced femoral neck fractures

The keywords are too descriptive, please shorten to 5-6 entries maximum. 

Response: We agree with you. Instruction for the author in this journal describes that “the keyword list three to ten”. We have revised the keywords as follows.

Line 38: femoral neck fracture; intracapsular hip fracture; Garden alignment index; posterior tilt; inter-rater reliability; intra-rater reliability; intraclass correlation coefficients

The introduction section is currently unsuitable for publication. It is firstly too generally written, and secondly it is rather a part of the introduction to another section-Discussion. Please enlarge this part of the paper to make it a real introduction to the topic of your research. 

Response: As suggested, we have revised the introduction and added citations as follows.

Line 44:

Recent systematic reviews and meta-analyses have revealed that preoperative posterior tilt ≥20° using lateral radiograph is associated with failed internal fixations [3,4]. On the other hand, preoperative anteroposterior (AP) alignment (valgus tilt > 15°) were the risk factors for failure in treatment [5]. In addition, the influence of postoperative AP alignment on reoperations or functional scores was evaluated recently [6,7].       

Line 63:

We believe that determining a more reliable measurement for AP alignment will contribute to better clinical decision-making and would serve as a base for extensive clinical research on femoral neck fractures.

Additional citations

[4] Papadelis, E.; Chaudhry, Y.P.; Hayes, H.; Talone, C.; Shah, M.P. Evaluation of the posterior tilt angle in predicting failure of nondisplaced femoral neck fractures after internal fixation: A systematic review [published online ahead of print, 2022 Sep 23]. J Orthop Trauma. 2022, DOI:10.1097/BOT.0000000000002490

[5] Song, H.K.; Choi, H.J.; Yang, K.H. Risk factors of avascular necrosis of the femoral head and fixation failure in patients with valgus angulated femoral neck fractures over the age of 50 years. Injury. 2016, 47, 2743-2748. DOI:10.1016/j.injury.2016.10.022

I would suggest replacing the phrase "junior and  senior surgeons" in the paper with surgeon with seniority.... and a surgeon with seniority.....

Response: We thank the reviewer for the suggestion. The terms, “senior surgeon” and “junior surgeon”, are commonly used in orthopedic papers [1,2,3]. However, we understand that the reviewer is concerned that the adjective, “senior”, may confuse readers into thinking that the senior surgeons were much older than the junior surgeons. For better clarification, we have explained what these terms mean in parentheses.

Line 98:

We defined the raters as senior surgeons (more experienced surgeons with 10 years of experience) or junior surgeons (less experienced surgeons with < 10 years of experience).  

[1] Jiamton C, Sayan P, Rungchamrussopa P, Kittithamvongs P. Traction-internal rotation radiograph can improve agreement in AO/OTA Classification System for intertrochanteric fracture. Indian J Orthop. 2022;56(11):1998-2005. Published 2022 Aug 23. doi:10.1007/s43465-022-00722-4

[2] Liu P, Xiao JX, Zhao C, et al. Factors associated with the accuracy of depth gauge measurements. Front Surg. 2022;8:774682. Published 2022 Jan 13. doi:10.3389/fsurg.2021.774682

[3] Ledwos N, Mirchi N, Bissonnette V, Winkler-Schwartz A, Yilmaz R, Del Maestro RF. Virtual reality anterior cervical discectomy and fusion simulation on the novel sim-ortho platform: Validation studies. Oper Neurosurg (Hagerstown). 2020;20(1):74-82. doi:10.1093/ons/opaa269

The paper should be enriched with more example photos. 

Response: As suggested, we have added three figures (radiograph showing a left nondisplaced femoral neck fracture, GAI measurement, and a difficult case of VTM measurement).

The Conclusion section should be completed, its current form is too laconic.

Response: As suggested, we have revised the conclusion and added these four points:

  1. Reliability was greater for GAI than for VTM in the study population.
  2. The reliability of GAI was influenced by patient age.
  3. GAI should be the first measurement choice.
  4. Future multicenter studies with more images and raters are necessary.

Line 267: This reliability analysis showed that although both GAI and VTM were reliable measurement methods, GAI was more reliable than VTM for assessing AP alignment using preoperative AP radiographs for nondisplaced femoral neck fractures. Additionally, the reliability of GAI was higher in patients aged < 80 years than in patients aged ≥ 80 years. Therefore, age-related variations in GAI measurement should be considered. Well-designed reliability studies with more images and raters are necessary to clarify the reliability of AP alignment assessment.

Reviewer 2 Report

The manuscript attempts to present a comparison of the reliability of the Garden alignment index (GAI) and a new anteroposterior (AP) alignment measurement (valgus tilt measurement [VTM]) using preoperative AP radiographs of nondisplaced femoral-neck fractures. The study was designed as an intra- and inter-rater reliability analysis with the main outcome like an intraclass correlation coefficient (ICC). The intra- and inter-rater reliability was calculated, using a mixed-effects model considering the rater, patient, and time. Results showed that GAI was a more reliable measurement method than VTM, using preoperative AP radiographs for nondisplaced femoral neck fractures, although both measurements were reliable. The reliabilities of GAI and VTM were higher for patients aged < 80 years than for those aged ≥ 80 years. No doubt, the paper is very interesting with different explanations and important roles, and it is the first study to evaluate the reliability of GAI assessment. Besides this, the authors described several limitations of this study too. The experience of the authors underlined that there wasn`t any information, related to this and other countries with different patient populations and image viewing systems well-designed studies. The authors indicate that more images and raters are necessary to clarify the reliability of AP alignment assessment in the future.

I would like to suggest just minor corrections from my side and only a few details I have underlined in the comments. Please, find the attached document describing a few of these minor issues noted in the manuscript.

Author Response

Response to Reviewer 2 Comments:

I would like to suggest just minor corrections from my side and only a few details I have underlined in the comments. Please, find the attached document describing a few of these minor issues noted in the manuscript.

Response: We thank Reviewer 2 for these comments.

Specific comments

Point 1: Please, look at the spaces between text on Page 4 of 8: the 2nd Paragraph looks different from other text in this section.

Point 2: Please, remove additional space between the word “doctors” and punctuation on Page 4 of 8: the 3rd Paragraph.

Response: We thank the reviewer for the suggestions. We have removed the additional spaces.

Round 2

Reviewer 1 Report

The authors followed the reviewer's suggestions. I have no further comments.